# No Free Lunch in Active Learning: LLM Embedding Quality Dictates Query Strategy Success

## Abstract

The advent of large language models (LLMs) capable of producing general-purpose representations lets us revisit the practicality of deep active learning (AL): By leveraging frozen LLM embeddings, we can mitigate the computational costs of iteratively fine-tuning large backbones. This study establishes a benchmark and systematically investigates the influence of LLM embedding quality on query strategies in deep AL. We employ five top-performing models from the massive text embedding benchmark (MTEB) leaderboard and two baselines for ten diverse text classification tasks. Our findings reveal key insights: First, initializing the labeled pool using diversity-based sampling synergizes with high-quality embeddings, boosting performance in early AL iterations. Second, the choice of the optimal query strategy is sensitive to embedding quality. While the computationally inexpensive Margin sampling can achieve performance spikes on specific datasets, we find that strategies like Badge exhibit greater robustness across tasks. Importantly, their effectiveness is often enhanced when paired with higher-quality embeddings. Our results emphasize the need for context-specific evaluation of AL strategies, as performance heavily depends on embedding quality and the target task.

## 1 Introduction

Self-supervised learning has transformed natural language processing (NLP), enabling models from BERT (Devlin et al., 2019) to large language models (LLMs) like GPT variants (Radford et al., 2018; 2019; Brown et al., 2020; OpenAI, 2023) to acquire powerful capabilities from vast unlabeled datasets. This paradigm shift, extending across domains like computer vision (Caron et al., 2021; Oquab et al., 2024), has yielded foundation models whose pre-trained embeddings often perform effectively without requiring full model fine-tuning (Oquab et al., 2024). Leveraging frozen embeddings offers an efficient pathway for deep learning research: it drastically reduces computational costs, circumvents complexities associated with fine-tuning LLMs (Rauch et al., 2023), and offers the ability to isolate the impact of the embedding itself. Since obtaining humanly labeled data for downstream tasks is still expensive, this renewed practicality invites a re-examination of computationally intensive methods to save annotation costs or those that require fast downstream adaptation, like deep active learning (AL). AL aims to maximize model performance under limited labeling budgets by selecting the most informative instances for annotation (Settles, 2009), a process reliant on embedding quality.

However, the utility of LLM embeddings is not uniform. Their quality, reflected by their ability to capture relevant semantic information, varies notably depending on the model architecture, pre-training data, and training objectives (Tao et al., 2024). Benchmarks like the massive text embedding benchmark (MTEB) (Muennighoff et al., 2023) aim to quantify this variability through evaluations on diverse static tasks (e.g., retrieval, classification) with a dynamic leaderboard.[1] However, how these static benchmark rankings translate to the effectiveness of embeddings within the dynamic, iterative instance selection process of AL remains largely unexplored. The effectiveness of AL query strategies critically depends on the underlying representation's ability to discern data characteristics (Hacohen et al., 2022). High-quality embeddings should theoretically enable AL strategies to identify ambiguous instances better or explore underrepresented data regions (Hacohen et al., 2022; Gupte et al., 2024). This dependency extends to the initial pool selection (IPS) in AL, where using

---

[1] https://huggingface.co/spaces/mteb/leaderboard_legacy

high-quality embeddings might offer significant advantages over the common practice of random initialization (Huseljic et al., 2024a; Gupte et al., 2024), especially in low-budget scenarios.

Despite the intuitive link between embedding quality and AL performance, this relationship remains unverified for LLMs as feature extractors. Deep AL research in NLP has focused on evaluating LLMs as labeling sources (Kholodna et al., 2024; Astorga et al., 2024) and comparing query strategies based on fine-tuning of small LMs (Rauch et al., 2023; Margatina et al., 2022; Schröder et al., 2022). This makes it challenging to isolate the contribution of the embedding quality from the fine-tuning dynamics, or to determine if a strategy's success is generalizable beyond specific training paradigms (Rauch et al., 2023). We raise the question: Is there a universally best query strategy, or does the optimal choice depend on the interplay between the embedding model, the query strategy, and the specific downstream task? This paper addresses this gap through a comprehensive benchmark study focused on frozen LLM embeddings within a practical deep AL framework, providing insights and reference performance data. Our contributions are:

---

**Contributions**

1. We conduct a comprehensive **benchmark study on how LLM embedding quality affects AL** by using five top-performing LLMs on the MTEB leaderboard and two baseline models on ten NLP tasks (Rauch et al., 2023) with seven query strategies.

2. We show that **diversity-based initial pool selection** (i.e., TypiClust) paired with strong embeddings yields a notable early-round advantage over random sampling.

3. We demonstrate that **AL strategy rankings vary with embedding model and task**. While Margin sampling is strong on specific datasets, Badge and Entropy demonstrate greater robustness and benefit from higher-quality embeddings.

4. We confirm that **no single strategy is universally superior**, exposing limitations of static benchmarks like MTEB for predicting AL utility and challenging future AL research to account for these contextual factors.

5. We release an **extensible deep AL framework**[a] built on the scikit-activeml package (Kottke et al., 2021), enabling reproducible experiments with any frozen LLM embeddings. This framework underpins our benchmark and fosters further research into practical deep AL pipelines.

---
[a]https://anonymous.4open.science/r/al-llm-embeddings

---

## 2 RELATED WORK

Starting from Table 1, which compares related AL benchmarks, this section discusses the key features of our benchmark in the broader context of AL and NLP research.

**Embeddings in NLP.** Large-scale pre-trained language models, enabled by transformers (Vaswani et al., 2017), have revolutionized NLP. Starting with encoder models like BERT (Devlin et al., 2019), the field has rapidly evolved towards LLMs, including decoder-centric architectures like the GPT series (Radford et al., 2018; 2019; Brown et al., 2020; OpenAI, 2023). The resulting robust, task-agnostic embeddings as frozen features bypass the need for full model fine-tuning (Oquab et al., 2024). For lightweight downstream tasks or settings with limited supervision, shallow model training (e.g., linear probing) can yield strong performance at a fraction of the cost of full fine-tuning. Moreover, many practical use cases, such as clustering or small-scale classification, do not require full-text generation, making generative LLMs unnecessarily expensive and often impractical at scale (Reimers & Gurevych, 2019). However, the quality of these embeddings varies across models, prompting benchmarks like MTEB (Muennighoff et al., 2023) for systematic evaluation. While MTEB provides valuable quality metrics, how this quality translates to dynamic, data-selection processes remains an open question central to our work.

**AL in NLP.** Deep AL is a well-studied paradigm across domains (Huseljic et al., 2024b; Rauch et al., 2023; 2024) aimed at reducing annotation costs by enabling a model to actively query labels for instances from which it expects to learn the most (Settles, 2009). Despite extensive research (Huseljic et al., 2024a), comparing the effectiveness of different AL query strategies has historically been difficult due to diverse experimental settings, particularly in the context of deep learning (Rauch et al.,

2023; Schröder et al., 2022; Margatina et al., 2022). Recognizing this challenge for transformer-based LMs, the ActiveGLAE benchmark (Rauch et al., 2023) provides a comprehensive collection of datasets (which we utilize in this study) and evaluation guidelines intended to facilitate and streamline the assessment of deep AL across studies. It identifies key challenges hindering comparisons, namely dataset selection, model training protocols, and AL settings. However, the initial baselines and focus within ActiveGLAE as well as previous research (Schröder et al., 2022; Margatina et al., 2022) primarily centered on encoder-only LMs like BERT (Devlin et al., 2019), evaluated using iterative fine-tuning throughout the AL process. While valuable for that specific setting, our work addresses the increasingly relevant paradigm of leveraging larger LLMs as fixed feature extractors, a computationally efficient approach common in transfer learning. This shift is also motivated by practical considerations: fine-tuning large generative LLMs within a typical classification-focused AL loop presents computational challenges. It requires complex model adaptations, making comparisons highly dependent on the specific fine-tuning protocol and the model itself. Utilizing frozen embeddings (i.e., linear probing) circumvents these issues, allowing us to isolate the impact of the initial embedding quality provided by LLMs, separate from the confounding factors of model fine-tuning. Within this frozen-embedding framework, we evaluate a range of established AL query strategies covering the main approaches (Zhan et al., 2022). For uncertainty sampling, which targets instances where the model is uncertain (Settles, 2009), we employ common methods like *Margin* sampling and *Entropy* sampling (Settles, 2009). We also investigate diversity-based sampling strategies, which aim to select instances representing the underlying data distribution, often by operating on the model's embeddings (Sener & Savarese, 2018). Specifically, we include *CoreSet* (Sener & Savarese, 2018), *ProbCover* (Yehuda et al., 2022), and *TypiClust* (Hacohen et al., 2022). Finally, we consider hybrid strategies that combine uncertainty and diversity principles, namely *BADGE* (Ash et al., 2020) and *DropQuery* (Gupte et al., 2024). By systematically evaluating these diverse strategies across different LLM embeddings of varying quality (as indicated by MTEB), we aim to understand how embedding characteristics influence strategy selection and performance in this practical frozen-feature setting.

**Initial pool selection in AL.** The AL process is iterative, requiring an initial pool of labeled data to train the first version of the model and enable uncertainty query strategies to operate, also known as the cold-start problem (Yuan et al., 2020). Recent research shows that the IPS is crucial for establishing a good classifier that works efficiently within AL and argues that IPS is an essential step in the AL cycle (Hacohen et al., 2022; Chen et al., 2024; Yehuda et al., 2022). Conventionally, the instances for this initial pool are selected randomly from the unlabeled dataset (Huseljic et al., 2024a; Rauch et al., 2023; Huseljic et al., 2024b). While straightforward, this random approach may not optimally represent the underlying data space effectively, leading to interest in exploring more informed selection strategies for the initial labeling phase.

Table 1: Comparison of **AL benchmarks**.

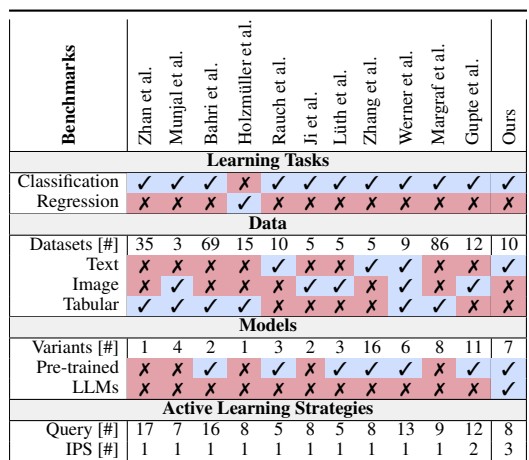

| Benchmarks | Zhan et al. | Munjal et al. | Bahri et al. | Holzmüller et al. | Rauch et al. | Ji et al. | Lüth et al. | Zhang et al. | Werner et al. | Margraf et al. | Gupte et al. | Ours |
|---|---|---|---|---|---|---|---|---|---|---|---|---|
| **Learning Tasks** | | | | | | | | | | | | |
| Classification | ✓ | ✓ | ✓ | ✗ | ✓ | ✓ | ✓ | ✓ | ✓ | ✓ | ✓ | ✓ |
| Regression | ✗ | ✗ | ✗ | ✓ | ✗ | ✗ | ✗ | ✗ | ✗ | ✗ | ✗ | ✗ |
| **Data** | | | | | | | | | | | | |
| Datasets [#] | 35 | 3 | 69 | 15 | 10 | 5 | 5 | 5 | 9 | 86 | 12 | 10 |
| Text | ✗ | ✗ | ✗ | ✗ | ✓ | ✗ | ✗ | ✓ | ✓ | ✗ | ✗ | ✓ |
| Image | ✗ | ✓ | ✗ | ✗ | ✗ | ✓ | ✓ | ✗ | ✓ | ✗ | ✓ | ✗ |
| Tabular | ✓ | ✓ | ✓ | ✓ | ✗ | ✗ | ✗ | ✗ | ✓ | ✓ | ✗ | ✗ |
| **Models** | | | | | | | | | | | | |
| Variants [#] | 1 | 4 | 2 | 1 | 3 | 2 | 3 | 16 | 6 | 8 | 11 | 7 |
| Pre-trained | ✗ | ✗ | ✓ | ✗ | ✓ | ✗ | ✓ | ✓ | ✗ | ✗ | ✓ | ✓ |
| LLMs | ✗ | ✗ | ✗ | ✗ | ✗ | ✗ | ✗ | ✗ | ✗ | ✗ | ✗ | ✓ |
| **Active Learning Strategies** | | | | | | | | | | | | |
| Query [#] | 17 | 7 | 16 | 8 | 5 | 8 | 5 | 8 | 13 | 9 | 12 | 8 |
| IPS [#] | 1 | 1 | 1 | 1 | 1 | 1 | 1 | 1 | 1 | 1 | 2 | 3 |

**AL with pretrained models.** The intersection of AL and pre-trained models is increasingly explored. One relevant line of research utilizes embeddings from pre-trained models as frozen features for AL. Notably, Gupte et al. (2024) demonstrate this approach using vision foundation models, highlighting the influence of rich representations on AL strategies, investigating IPS, and introducing the Drop-Query strategy. Our work builds directly on this paradigm but shifts the focus systematically to the NLP domain, examining a diverse set of modern LLM embeddings and their quality's impact on various AL strategies and IPS effectiveness. Other studies have investigated AL strategy performance in different contexts, such as Bahri et al. (2022), who found Margin sampling effective using pre-trained models on tabular data. However, findings may not directly transfer across domains. In contrast to the frozen-feature approach, another research direction involves iteratively fine-tuning pre-trained models during the AL cycle (Hacohen et al., 2022; Tamkin et al., 2022). While showing benefits over training from scratch, this differs significantly from our setting regarding computational cost and the goal of isolating the intrinsic contribution of the initial embedding quality. Our work adopts the computationally efficient frozen setting common in transfer learning scenarios.

## 3 EXPERIMENTAL SETUP

Our experimental setup targets studying the interplay between LLM embedding quality and deep AL.

**Problem setting.** We implement an AL cycle where pre-trained LLMs serve as frozen feature extractors. This follows the trend in AL for computer vision (Gupte et al., 2024; Hacohen et al., 2022). Let $h_\omega : \mathcal{X} \to \mathbb{R}^D$ denote the pre-trained LLM with frozen parameters $\omega$, mapping an input text $x \in \mathcal{X}$ to a $D$-dimensional embedding $h_\omega(x)$. A classification head then processes this embedding, $f_{\theta_t} : \mathbb{R}^D \to \mathbb{R}^C$, where $C$ is the number of classes. For linear probing, $f_{\theta_t}$ is a linear classifier with parameters $\theta_t$ at AL cycle iteration $t$. It predicts class probabilities $\hat{p} = \sigma(f_{\theta_t}(h_\omega(x)))$, where $\sigma$ denotes the softmax function. This frozen-feature extractor setup offers two main advantages: First, input embeddings $h_\omega(x)$ only need to be computed once for all instances and can be reused throughout the AL process, yielding notable computational savings. Second, this configuration effectively isolates the contribution of the embedding quality (provided by $h_\omega$) to the AL performance, allowing a precise assessment across models without the confounding effects of model fine-tuning. Our experiments operate within a pool-based AL classification setting, starting with a large unlabeled pool $\mathcal{U}^{(0)} \subseteq \mathcal{D}$ (where $\mathcal{D}$ is the entire dataset) and an initially labeled pool $\mathcal{L}^{(0)}$ of size $k_0 = |\mathcal{L}^{(0)}|$.

**Data, models, and training.** To assess the impact of LLM embeddings in deep AL, our experiments utilize the ActiveGLAE benchmark (Rauch et al., 2023). This benchmark provides a standardized set of ten diverse NLP classification tasks, chosen to include variations in label imbalance, number of classes, and domain focus. This variety makes the datasets well-suited for a meaningful evaluation across different scenarios. Dataset specifics are detailed in Table 2. We select five state-of-the-art LLM embedding models, chosen as top performers on the MTEB leaderboard (Muennighoff et al., 2023). We compare them against two baselines: BERT (Devlin et al., 2019), prevalent in prior AL research for NLP (Rauch et al., 2023; Schröder et al., 2022), and its successor, MODERN-BERT (Warner et al., 2024). For each model, we obtain embeddings by pooling the appropriate token (`[CLS]` or `[EOS]`) according to its specification. Detailed information on these models, including pooling strategy, parameter count, embedding dimension, MTEB score (the mean score determining leaderboard rank), and ranking, is provided in Table 3. We repeat all experiments five times using distinct random seeds. Results are presented as mean and standard deviation. For direct comparisons (e.g., evaluating the difference between strategies, models, or cycle performance), we employed paired analyses at the seed level (comparing seed $i$ results only with other seed $i$ results). Embeddings serve as inputs to the linear classifier in the form of a logistic regression model (Pedregosa et al., 2011), which is optimized with the LBFGS solver for up to 1,000 iterations.

Table 2: **Datasets** from ACTIVEGLAE.

| Name | \|Train\| | \|Test\| | # Classes | Budget |
|---|---|---|---|---|
| AG's News | 120k | 7600 | 4 | 1000 |
| Banking77 | 10k | 3000 | 77 | 5000 |
| DBPedia | 560k | 5000 | 14 | 500 |
| FNC-1 | 40k | 4998 | 4 | 3500 |
| MNLI | 390k | 9815 | 3 | 4000 |
| QNLI | 104k | 5463 | 2 | 4500 |
| SST-2 | 67k | 872 | 2 | 500 |
| TREC-6 | 5452 | 500 | 6 | 1000 |
| Wiki Talk | 159k | 64k | 2 | 3000 |
| Yelp-5 | 650k | 50k | 5 | 2500 |

Table 3: **Models** from the MTEB leaderboard.[2]

| Name | # Params. | # Dim. | Pooling | MTEB Pos | MTEB Score |
|---|---|---|---|---|---|
| NVIDIA-EMBED-V2 | 7.8B | 4096 | CLS | 1 | 72.31 |
| BGE-EN-ICL | 7.1B | 4096 | EOS | 2 | 71.67 |
| STELLA-V5 | 1.5B | 1536 | CLS | 3 | 71.19 |
| SFR-EMBED-2 | 7.1B | 4096 | EOS | 4 | 70.31 |
| QWEN2.5-7B | 7.6B | 3584 | EOS | 6 | 69.59 |
| MODERNBERT-BASE | 150M | 768 | CLS | – | 62.61 |
| BERT-BASE-UNCASED | 110M | 768 | CLS | 174 | 38.33 |

**Initial pool selection.** Since IPS methods are influenced by the initial pool size $k_0$ (Hacohen et al., 2022; Gupte et al., 2024), we first conduct experiments focused explicitly on this stage. We evaluate two diversity-based strategies (CoreSet, TypiClust) and random sampling for selecting an initial labeled pool $\mathcal{L}^{(0)}$ of varying sizes (ranging from 50 to 5,000 instances) from $\mathcal{U}^{(0)}$. To isolate the impact of the IPS strategy and size, we train the classifier $f_{\theta_1}$ only on the initial pool $\mathcal{L}^{(0)}$ and evaluate its initial performance. The best-performing IPS strategy in this evaluation is then used throughout the main AL cycle experiments.

**AL cycle.** For the AL experiments evaluating query strategies over multiple iterations, we select an initial labeled pool $\mathcal{L}^{(0)}$. The process then iterates for $T = 20$ cycles. In each cycle $t = 0, \ldots, T-1$:

---

[2]Snapshot was taken in December 2024. We use GWEN2.5 since the 5th model was another STELLA model.

1. A query strategy selects a batch $\mathcal{B}^{(t)} \subset \mathcal{U}^{(t)}$ of size $|\mathcal{B}^{(t)}| = b$ instances that it expects to yield the highest performance gains.

2. These instances are annotated, yielding $\mathcal{B}^{*(t)} \subset \mathcal{D} \times \mathcal{Y}$.

3. The pools are updated: $\mathcal{U}^{(t+1)} = \mathcal{U}^{(t)} \setminus \mathcal{B}^{(t)}$ and $\mathcal{L}^{(t+1)} = \mathcal{L}^{(t)} \cup \mathcal{B}^{*(t)}$.

4. The logistic regression classifier $f_{\theta_t}$ is retrained from scratch on the entire updated labeled pool $\mathcal{L}^{(t+1)}$, resulting in updated parameters $\theta_{t+1}$.

While the number of cycle iterations is fixed, the total labeling budget $B = |\mathcal{L}^{(0)}| + T \times b$ varies per dataset. We empirically determine this budget $B$ by observing a baseline configuration's performance convergence point on each dataset (BERT embeddings with random sampling). Table 2 details the resulting dataset-specific budget sizes. After the IPS, the fixed total budget $B$ and the number of cycles $T = 20$ determine the batch size $b$ for each AL iteration.

## 4 BENCHMARK RESULTS

Our benchmark investigation focuses on IPS and subsequent AL cycles under varying embedding qualities. It yields three main findings that provide a performance landscape for this setting: (1) While not all diversity-based IPS strategies perform well, TypiClust, which selects diverse and representative instances from the embedding space, provides notable advantages over random sampling when coupled with high-quality embeddings, particularly in early AL cycles or low-budget scenarios. (2) Uncertainty-based (Margin, Entropy) and hybrid (Badge) strategies exhibit robust performance across varying embedding qualities and tasks during the AL cycles. (3) Embedding quality directly impacts model performance within the AL cycle, as superior representations accelerate the transition from initial diversity-based exploration to effective uncertainty-based exploitation later in the cycle.

### 4.1 EFFECTIVENESS OF INFORMED INITIAL POOL SELECTION

This section compares informed IPS strategies to random initialization. In particular, we focus on the interplay between IPS strategy, initial pool size ($k_0$), dataset characteristics, and embedding quality. We compare three IPS strategies: Random sampling (baseline), CoreSet (diversity), and TypiClust (diversity and representativeness) across gradually increasing initial pool sizes, ranging from $k_0 = 20$ to $k_0 = 5000$ instances, using the performance achieved after training only on this initial pool.

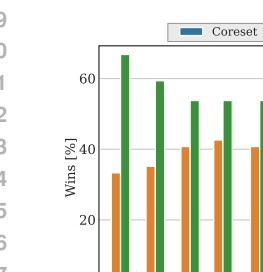

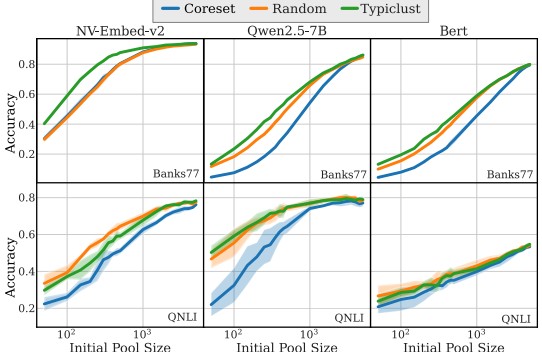

Figure 1: **Winning IPS strategy frequency** (counts), aggregated across 10 datasets and 7 embedding models for varying initial pool sizes ($k_0$). Ties are excluded.

Figure 2: **IPS performance comparison** (accuracy vs. initial pool size $k_0$) for selected embedding models on Banks77 (top) and QNLI (bottom) datasets.

**Aggregated results.** Figure 1 aggregates the results, showing which strategy performs best most frequently across all 10 datasets and 7 embedding models for each initial pool size. TypiClust, which aims to select diverse and representative instances through clustering, outperforms random sampling and CoreSet for smaller initial pools, particularly those up to approximately 250-300 instances. In contrast, CoreSet, which focuses on geometric coverage of the embedding space, consistently performs worse than the random baseline, suggesting its diversity approach is unsuitable

for informed IPS in our frozen-feature setting. This also confirms previous results for general AL performance (Rauch et al., 2023). As the initial pool size ($k_0$) grows beyond 300 instances, the advantage of TypiClust diminishes, and random sampling becomes competitive. This indicates that once a sufficiently large initial set is selected, it captures enough diversity for effective initial training.

**Impact of task.** The overall trend masks important details of datasets and embedding models, as illustrated in Figure 2. This figure shows performance curves for selected models (BERT, QWEN2.5, NV-EMBED-V2) on two contrasting datasets: Banks77 (highly multi-class, fine-grained) and QNLI (binary classification). On Banks77, the advantage of TypiClust is pronounced and persists even with larger initial pools (e.g., $k_0 > 500$ for high-quality embeddings like NV-EMBED-V2). This highlights the value of targeting representativeness via TypiClust when the task involves distinguishing between many fine-grained classes. On QNLI, the performance differences between TypiClust and random sampling are marginal, even for tiny initial pools ($k_0 = 50$). This suggests that the benefit of informed IPS strategies is limited for simpler tasks where classes might be more easily separable in the embedding space. These examples underscore that the effectiveness of an IPS strategy is context-dependent, influenced by factors like dataset complexity and the number of classes.

**Impact of embedding model.** The quality of the underlying embedding model also modulates the effectiveness of diversity-based IPS. As seen in Figure 2 (comparing BERT vs. NV-EMBED-V2 on Banks77), higher-quality embeddings enable TypiClust to achieve a greater performance advantage over random sampling, and this advantage often persists across a broader range of initial pool sizes ($k_0$). This suggests a synergy: superior embeddings provide a richer, more discriminative feature space where diversity and representativeness can more effectively identify a varied and informative initial set of instances, leading to a better starting model. While not a perfect correlation across every model-dataset pair, there is a general tendency for embeddings ranked higher on MTEB to derive greater benefit from TypiClust initialization, especially when the initial budget ($k_0$) is small.

> **Takeaway: Effectiveness of IPS**
>
> The benefit of informed IPS depends on the context. The diversity-based strategy TypiClust provides notable advantages over random sampling, primarily for small initial labeled pools ($k_0 < 300$) and high-quality embeddings (according to MTEB ranks), especially on complex datasets. The advantage diminishes with larger initial pools or simpler tasks. CoreSet consistently underperforms random sampling for IPS in this setting. Choosing an IPS strategy requires considering the available budget, dataset complexity, and quality of the embeddings.

## 4.2 IMPACT OF INFORMED INITIAL POOL SELECTION ON ACTIVE LEARNING

Based on our findings in the previous section, where TypiClust demonstrated superior IPS performance over CoreSet and random sampling, we now investigate how the use of TypiClust versus random sampling influences the performance dynamics over subsequent AL cycles.

**Impact of task and embedding model.** Figure 3 illustrates the impact of the IPS choice by plotting the performance difference (accuracy of strategy with TypiClust IPS minus accuracy of the same strategy with random IPS) over the AL cycle for various query strategies. The plots again show results for selected embedding

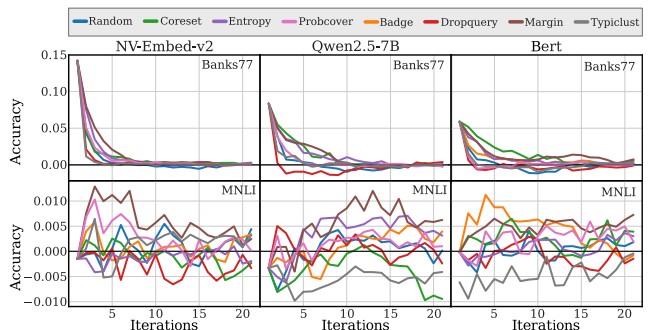

Figure 3: Performance difference (TypiClust IPS vs. random IPS) over AL cycles for query strategies. Rows correspond to datasets, columns to selected embedding models. Positive values indicate that TypiClust IPS is better than random.

models (BERT, QWEN2.5, NV-EMBED-V2) on two different datasets (Banks77 and MNLI). A consistent pattern emerges across many dataset-model combinations: TypiClust IPS provides an initial performance advantage (positive difference) in the early AL iterations. This head start is particularly pronounced for complex, multi-class datasets like Banks77, where the initial performance gain can be substantial (e.g., differences exceeding 5-10 percentage points for NV-EMBED-V2 in the very

first cycles, as seen in Figure 3). This suggests that starting with a diverse set selected by TypiClust allows the model to learn a better initial decision boundary. However, the performance difference generally diminishes and approaches zero within approximately 5 to 10 AL cycles, indicating that the models initialized randomly eventually catch up as more informative instances are acquired through selected AL query strategies. For simpler tasks like MNLI, the initial advantage from TypiClust is less pronounced and more variable across strategies and models. Additionally, the embedding quality (i.e., the ranking in MTEB) also strongly influences the performance gains: The NV-EMBED-V2 model shows stronger improvements compared to lower ranks. A comprehensive overview of results across all datasets and models is available in Appendix B.

**Influence on query strategy.** Beyond the initial performance boost, the choice of IPS influences the relative effectiveness of query strategies during the subsequent AL cycles, as shown by the pairwise win rate matrices in Figure 4. These matrices, aggregated across datasets, models, and cycles, reveal that the overall hierarchy of query strategies remains largely consistent: Badge, Entropy, and Margin consistently achieve high average win rates (0.62-0.74), while CoreSet remains the weakest (0.15 mean win rate), regardless of the IPS method. However, comparing the two figures highlights nuanced shifts. Top-performing strategies (Badge, Entropy, Margin) exhibit slightly higher mean win rates when initialized with TypiClust, suggesting the diverse start allows them to capitalize more effectively. Specifically, the comparison shows that the uncertainty-based strategies, Entropy and Margin, slightly improve their relative advantage over continued diversity-based sampling (i.e., using TypiClust as a query strategy) when the AL process is initiated with TypiClust IPS. For example, Entropy's win rate against TypiClust increases from 0.69 to 0.73. This supports the intuition that TypiClust IPS ensures initial diversity: the AL process benefits more readily from transitioning to uncertainty sampling to refine the decision boundary, slightly diminishing the relative value of further diversity exploration via the TypiClust query strategy itself. Thus, while not significantly altering the strategy win rates, TypiClust IPS appears to precondition the learning process, subtly favoring subsequent uncertainty-based or hybrid strategies like Badge.

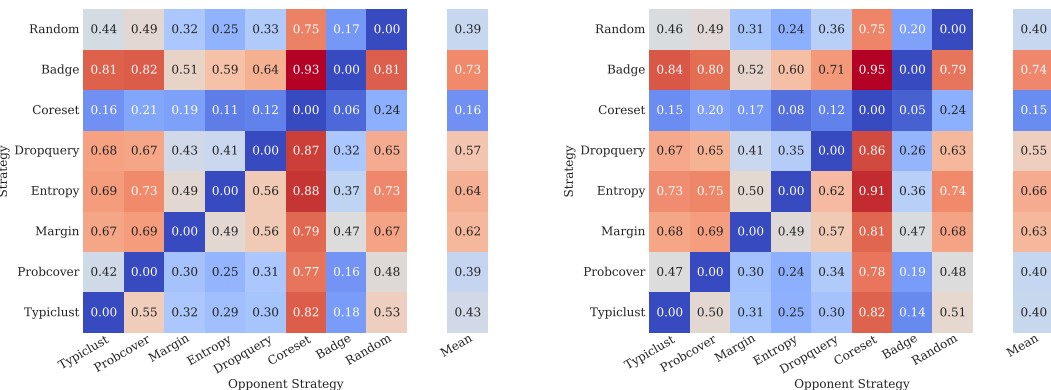

Figure 4: Strategies' pairwise win rates for **random** (left) and **TypiClust** (right) IPS. Values indicate the percentage of dataset-model-cycle combinations where the strategy outperforms its opponent.

> **Takeaway: Impact of IPS on AL**
>
> TypiClust IPS often increases early performance, particularly for complex tasks. This increase is larger for higher-quality embeddings (e.g., NV-EMBED-V2) but typically diminishes over 5-10 AL cycles. While the overall ranking of query strategies remains stable regardless of IPS, TypiClust initialization enhances the effectiveness of top strategies (Badge, Entropy, Margin). Specifically, it slightly increases the relative effectiveness of subsequent strategies prioritizing uncertainty over continued diversity-based sampling, potentially allowing models trained on better embeddings to transition faster towards refining decision boundaries (exploitation).

### 4.3 IMPACT OF EMBEDDING QUALITY ON ACTIVE LEARNING

This section analyzes the performance of various query strategies when used with embeddings from different LLMs, ranked according to their MTEB scores (cf. Tab. 3). We aim to demonstrate the role of embedding quality in the AL process by examining performance across multiple datasets and query strategies. We conduct all experiments with TypiClust as our best-performing IPS strategy.

**General results.** A primary observation is that higher-quality embeddings from LLMs enhance the downstream classification performance in AL compared to the baseline embeddings, like those from BERT. Figure 6 illustrates this at the example of the Banks77 and MNLI datasets, where using embeddings from top-ranked models like NV-EMBED-V2 not only leads to higher overall accuracy but can also enable faster convergence within the fixed budget compared to BERT embeddings. The specific strategy performances indicate that Margin and Badge are frequently the best-performing strategies across all components, including datasets, embedding models, and AL cycle iterations. We highlight this by showcasing how these strategies perform by isolating each component.

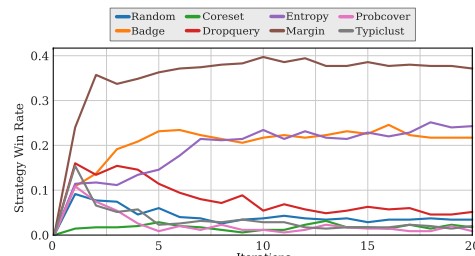

Figure 5: Win rate of each strategy being the top performer per **AL cycle**, **aggregated across all embeddings and datasets**. Ties are excluded.

**Impact of cycle iteration.** Figure 5 tracks the prevalence of the top-performing strategy across AL cycles, aggregated over all datasets and embeddings. Margin sampling is superior after the initial cycle(s), holding the top spot in over 35% of cases throughout the process. However, Entropy and Badge also demonstrate sustained high performance, each securing the top spot in roughly 20% of cases after the fifth cycle. This highlights the persistent value of uncertainty (Entropy, Margin) and hybrid (Badge) based strategies, while diversity-based strategies like DropQuery show diminishing returns over time. This figure visualizes shifting from diversity-centric exploration towards uncertainty exploitation as the AL process matures (Hacohen et al., 2022).

**Impact of embedding source.** The influence of the specific embedding source is highlighted in Figure 7 (left), which shows top strategy prevalence broken down by embedding model, aggregated over datasets and cycles. Margin sampling exhibits robustness, performing well across the entire spectrum of embedding qualities, from baseline BERT embeddings to high-ranked LLM embeddings like NV-EMBED-V2. In contrast, the effectiveness of Badge is more strongly influenced by the quality of the input embeddings. Badge achieves notably higher top-performance rates when utilizing higher-quality LLM embeddings compared to the baseline BERT embeddings (with STELLA being an exception). This suggests that Badge's hybrid nature makes it particularly dependent on leveraging the richer information present in better representations.

**Impact of classification task.** Figure 7 (right) underscores the task dependency of strategies. Margin sampling works very well on simpler datasets (e.g., AGNews, DBPedia, TREC6) with winning rates from 76 to 94%. However, on more complex datasets (e.g., Banks77, Yelp5, MNLI), Entropy and Badge emerge as the winners, with Margin showing little success with low win rates. This dataset-specific behavior, coupled with the higher average pairwise win rates observed for Badge and Entropy (Figure 4), suggests that while Margin can be effective for less complex tasks, Badge and Entropy offer greater overall robustness across diverse datasets and embedding types.

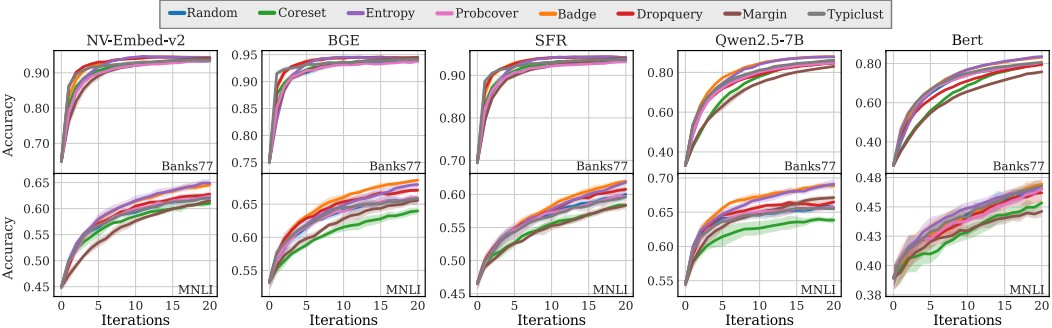

Figure 6: Performance comparison using NV-EMBED-V2, BGE, SFR, QWEN2.5, and BERT embeddings on Banks77 (top) and MNLI (bottom) datasets over AL cycles with **random** IPS.

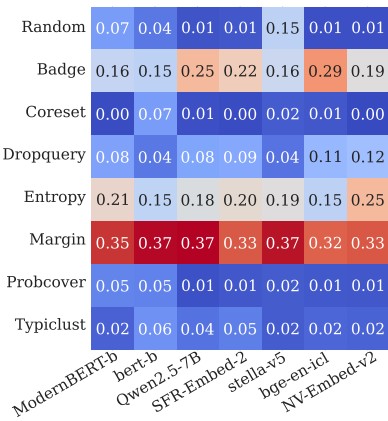
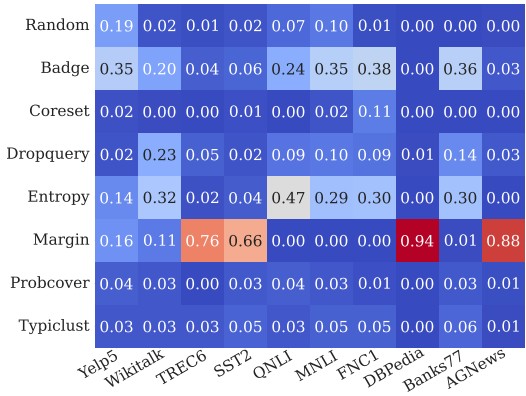

Figure 7: Win rates of being the best strategy per **embedding model** (left grid) and per **dataset** (right grid), **aggregated across all datasets and AL cycles**. Ties are excluded.

---

**Takeaway: Impact of embedding quality on AL**

High-quality LLM embeddings generally boost AL performance compared to baseline embeddings. Margin sampling exhibits strong performance spikes and is relatively robust to varying embedding qualities. Badge and Entropy emerge as more consistently high-performing and robust strategies across datasets and AL cycles. Notably, Badge's effectiveness is enhanced by higher-quality LLM embeddings. This reinforces that uncertainty-based and hybrid strategies are adept at leveraging informative embeddings, especially as the AL process matures and shifts from exploration (diversity) to exploitation (uncertainty). The best strategy choice remains context-dependent, necessitating consideration of the embedding source and task characteristics.

---

## 5 CONCLUSION AND LIMITATIONS

**Conclusion.** This work presents a benchmark study that systematically investigated the influence of frozen LLM embedding quality in deep AL across a diverse set of text classification tasks. Our results also serve as a reference for future research across domains. We employed a practical frozen feature setting while leveraging top-ranked MTEB embedding models and current AL strategies. We examined the interplay between embedding quality, IPS, query strategy performance, and task differences. We demonstrate that choosing an IPS strategy interacts with embedding quality. Concretely, informed IPS via TypiClust provides a performance advantage in early AL cycles compared to random IPS, particularly with high-quality embeddings on complex datasets. However, this initial head start typically diminishes after a few AL cycles. Analyzing the query strategies, our results revealed nuanced performances: While Margin sampling demonstrates strong efficacy on specific tasks, strategies like Badge show greater overall robustness across tasks. The effectiveness of Badge is often amplified when paired with higher-quality embeddings. This underscores that the optimal query strategy is highly context-dependent, influenced by both the embedding model and the specific task characteristics, further challenging the notion of a universally superior AL strategy. We also observed that using TypiClust IPS subtly accelerated the transition where uncertainty-based strategies start to outperform diversity-based ones in subsequent AL iterations. Furthermore, we reaffirmed the impact of overall embedding quality: models with higher MTEB rankings generally facilitate superior performance and faster convergence within the AL cycle.

**Limitations.** Our focus on frozen embeddings with a logistic regression classifier, while allowing for precise analysis of embedding impact, does not explore the dynamics of fully fine-tuning LLMs within the AL loop. Additionally, while diverse, the evaluation across ten NLP classification tasks may not cover all possible scenarios, and the insights derived from MTEB (primarily a dynamic retrieval-focused benchmark) might not fully generalize to all AL applications across domains. Despite these limitations, this study highlights the complex interplay between embedding quality, initial pool selection, query strategy selection, and task specifics in designing practical and efficient deep AL pipelines using LLM embeddings.

## ETHICS STATEMENT

This work uses publicly available datasets from the ActiveGLAE benchmark, which may contain societal biases (e.g., in representation or labeling). Our active learning framework does not mitigate these biases and could potentially amplify them through query selection. The pre-trained embedding models may propagate training data biases, and our frozen-feature setup promotes efficiency but limits debiasing. We encourage responsible use of our released framework, especially in sensitive domains.

## REPRODUCIBILITY STATEMENT

Our experiments are reproducible via the released extensible framework (built on scikit-activeml (Kottke et al., 2021)) at `https://anonymous.4open.science/r/al-llm-embeddings`, which includes all code, hyperparameters, and setup instructions. We use datasets from ActiveGLAE (Table 2), specified MTEB models (Table 3), and logistic regression via scikit-learn. All runs (5 seeds per configuration) were conducted on a NVIDIA A100 GPU for embeddings and AMD EPYC CPUs for active learning experiments. Further details about the total CPU runtime can be found in Appendix 5). Embeddings are pre-computed and reused for consistency and efficiency.

## LLM USAGE STATEMENT

LLMs were used solely as writing assistants for improvements to tables, grammar, wording, and clarity in the manuscript. They were also employed for light code debugging to identify and fix minor coding tasks, such as syntax or logical errors. All core research ideas, experimental design, primary code implementation, data analysis, and benchmark results were developed independently by the authors without LLM involvement. This limited use aligns with our commitment to original authorship and responsible AI practices.

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

## A    COMPUTATIONAL RESOURCES AND EXPERIMENT RUNTIME

This appendix provides the computational resources and experiment runtimes of the benchmark. The initial computation of LLM embeddings utilized an NVIDIA A100 GPU (80GB VRAM) on a SLURM cluster. All further experimental steps, including the iterative training of the logistic regression classifier and the execution of active learning query strategies (some of which, e.g., BADGE, are memory-intensive), were carried out on AMD EPYC CPUs equipped with up to 80GB RAM. Table 4 details the computation time required to generate embeddings for all datasets using a single A100 GPU.

Table 4: Total Runtime to create embeddings for all datasets.

| Model | Runtime (dd-hh:mm) |
|---|---|
| NVIDIA-EMBED-V2 (Lee et al., 2025) | 01-18:26 |
| BGE-EN-ICL (Xiao et al., 2024) | 03-09:16 |
| STELLA-V5 Zhang et al. (2025) | 00-22:06 |
| SFR-EMBED-2 (Meng et al., 2024) | 03-09:14 |
| QWEN2.7-7B (Meng et al., 2024) | 02-19:48 |
| MODERNBERT Warner et al. (2024) | 00-09:15 |
| BERT-BASE-UNCASED (Devlin et al., 2019) | 00-03:16 |

Table 5 presents the average runtime and standard deviation over 5 seeded runs for each strategy and dataset combination, exemplified by the **random** IPS strategy across all models. The total CPU time for our AL experiments using **random** IPS was approximately 30 days and 16 hours, while experiments with **TypiClust** IPS took around 33 days and 3 hours.

Table 5: Average experiment runtime (mm:ss) for all strategy and dataset combinations of **random** IPS across all models.

| Strategy | Dataset | Runtime (mm:ss) | Strategy | Dataset | Runtime (mm:ss) | Strategy | Dataset | Runtime (mm:ss) |
|---|---|---|---|---|---|---|---|---|
| CoreSet | AGNews | 02:17±01:10 | Badge | AGNews | 04:43±03:18 | Random | AGNews | 00:44±00:16 |
| | Banks77 | 15:25±06:53 | | Banks77 | 22:50±33:20 | | Banks77 | 08:07±04:00 |
| | DBPedia | 01:55±00:55 | | DBPedia | 07:34±05:35 | | DBPedia | 01:16±00:32 |
| | FNC1 | 08:21±05:09 | | FNC1 | 20:16±15:50 | | FNC1 | 03:19±03:16 |
| | MNLI | 09:52±06:19 | | MNLI | 15:14±11:24 | | MNLI | 04:32±03:47 |
| | QNLI | 09:09±05:33 | | QNLI | 11:17±08:30 | | QNLI | 01:15±00:43 |
| | SST2 | 01:17±00:37 | | SST2 | 01:36±00:59 | | SST2 | 00:25±00:05 |
| | TREC6 | 02:09±00:54 | | TREC6 | 04:02±02:36 | | TREC6 | 00:39±00:13 |
| | Wikitalk | 04:57±02:32 | | Wikitalk | 07:20±05:11 | | Wikitalk | 00:54±00:21 |
| | Yelp5 | 05:54±03:33 | | Yelp5 | 12:09±07:32 | | Yelp5 | 03:12±01:46 |
| DropQuery | AGNews | 06:20±03:08 | Entropy | AGNews | 01:16±00:36 | TypiClust | AGNews | 22:57±12:48 |
| | Banks77 | 23:08±10:22 | | Banks77 | 18:19±10:22 | | Banks77 | 55:36±06:29 |
| | DBPedia | 06:59±03:44 | | DBPedia | 01:37±00:47 | | DBPedia | 11:55±06:37 |
| | FNC1 | 13:06±09:04 | | FNC1 | 07:04±05:41 | | FNC1 | 32:34±52:49 |
| | MNLI | 16:49±10:52 | | MNLI | 07:40±06:09 | | MNLI | 51:58±09:13 |
| | QNLI | 08:23±04:46 | | QNLI | 02:22±01:52 | | QNLI | 01:37±06:44 |
| | SST2 | 05:33±02:56 | | SST2 | 00:30±00:06 | | SST2 | 11:47±06:31 |
| | TREC6 | 03:48±01:45 | | TREC6 | 01:11±00:34 | | TREC6 | 10:58±05:27 |
| | Wikitalk | 06:17±03:15 | | Wikitalk | 01:54±01:02 | | Wikitalk | 14:20±42:36 |
| | Yelp5 | 11:39±06:04 | | Yelp5 | 05:15±03:25 | | Yelp5 | 03:50±36:56 |
| ProbCover | AGNews | 03:16±00:51 | Margin | AGNews | 02:09±01:24 | | | |
| | Banks77 | 16:19±06:06 | | Banks77 | 08:35±04:26 | | | |
| | DBPedia | 03:33±01:23 | | DBPedia | 03:04±02:04 | | | |
| | FNC1 | 09:35±04:18 | | FNC1 | 04:13±03:44 | | | |
| | MNLI | 11:03±04:11 | | MNLI | 03:05±01:57 | | | |
| | QNLI | 09:31±02:09 | | QNLI | 01:01±00:35 | | | |
| | SST2 | 02:20±00:37 | | SST2 | 01:04±00:38 | | | |
| | TREC6 | 01:31±00:28 | | TREC6 | 02:10±01:15 | | | |
| | Wikitalk | 05:51±00:42 | | Wikitalk | 01:21±00:38 | | | |
| | Yelp5 | 07:37±02:47 | | Yelp5 | 04:07±02:40 | | | |

# B  ADDITIONAL RESULTS

This appendix provides supplementary figures that offer a more comprehensive visualization of the deep AL benchmark experiments discussed in the main paper. To facilitate a broad visual comparison across different embedding models and datasets, we utilize large matrix plots. While this makes individual subplots smaller, it enables a quick assessment of general trends.

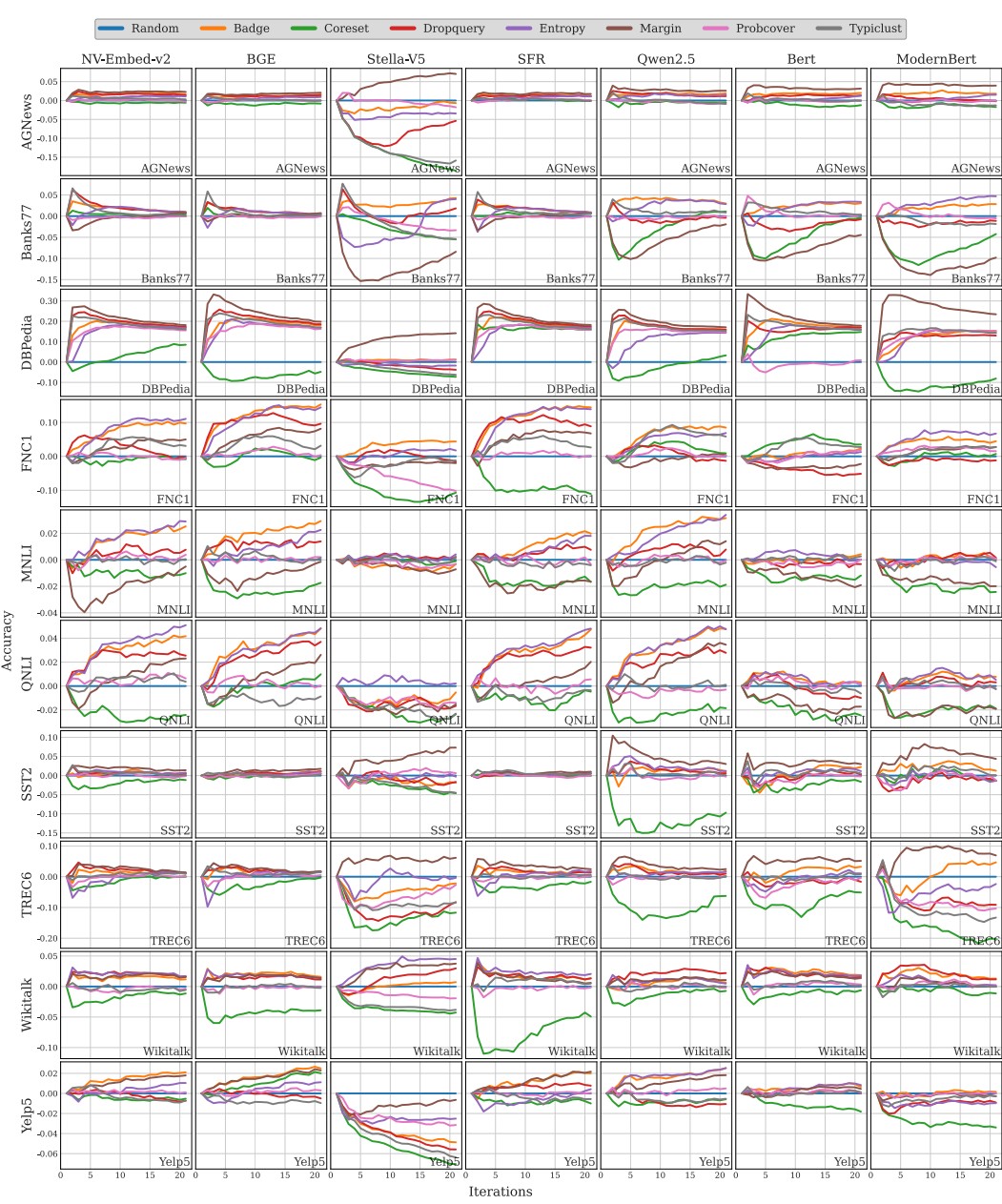

Figure 8: Performance difference (TypiClust IPS vs. random IPS) over AL cycles (0-20). Rows correspond to datasets, columns to embedding models. Positive values indicate that TypiClust IPS is better than random IPS.

Specifically, Figure 8 serves as a direct extension to Figure 3 from the main text. Whereas the main paper figure selectively illustrated the performance difference between TypiClust IPS and random IPS for three embedding models (NV-EMBED-V2, QWEN2.5B, BERT) on two datasets (Banks77, MNLI), this appendix figure presents the complete set of these difference plots across all

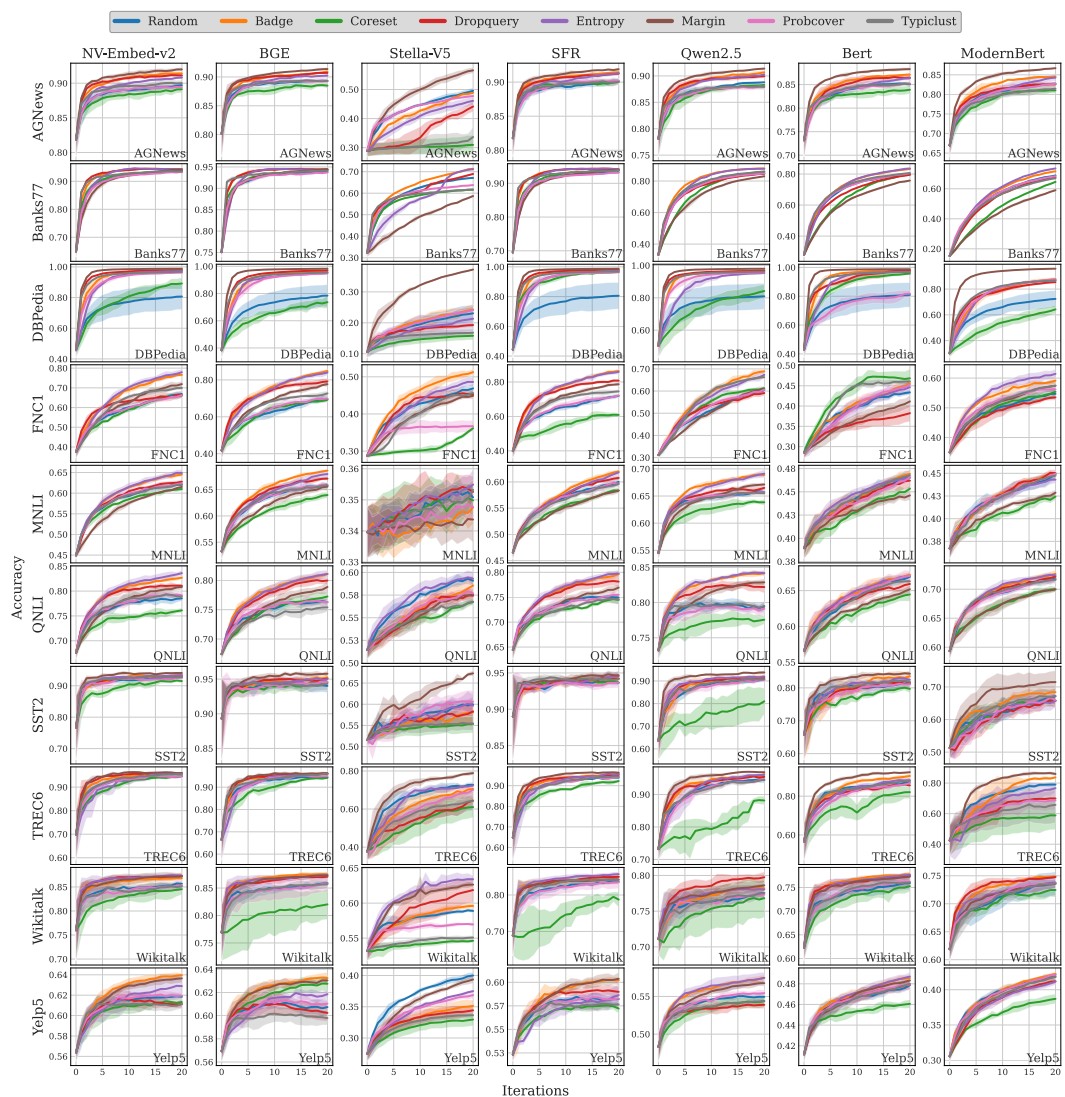

Figure 9: Performance comparison using of models and datasets over AL cycles with **Random** IPS.

model-dataset combinations investigated. Furthermore, Figures 9 and 10 deliver a complete overview of the AL performance curves for all experiments.

Figure 9 details the results when using the random IPS strategy, and Figure 10 provides the corresponding comprehensive results for experiments initiated with TypiClust as the IPS strategy. These figures show performance across all AL cycles, embedding models, and datasets.

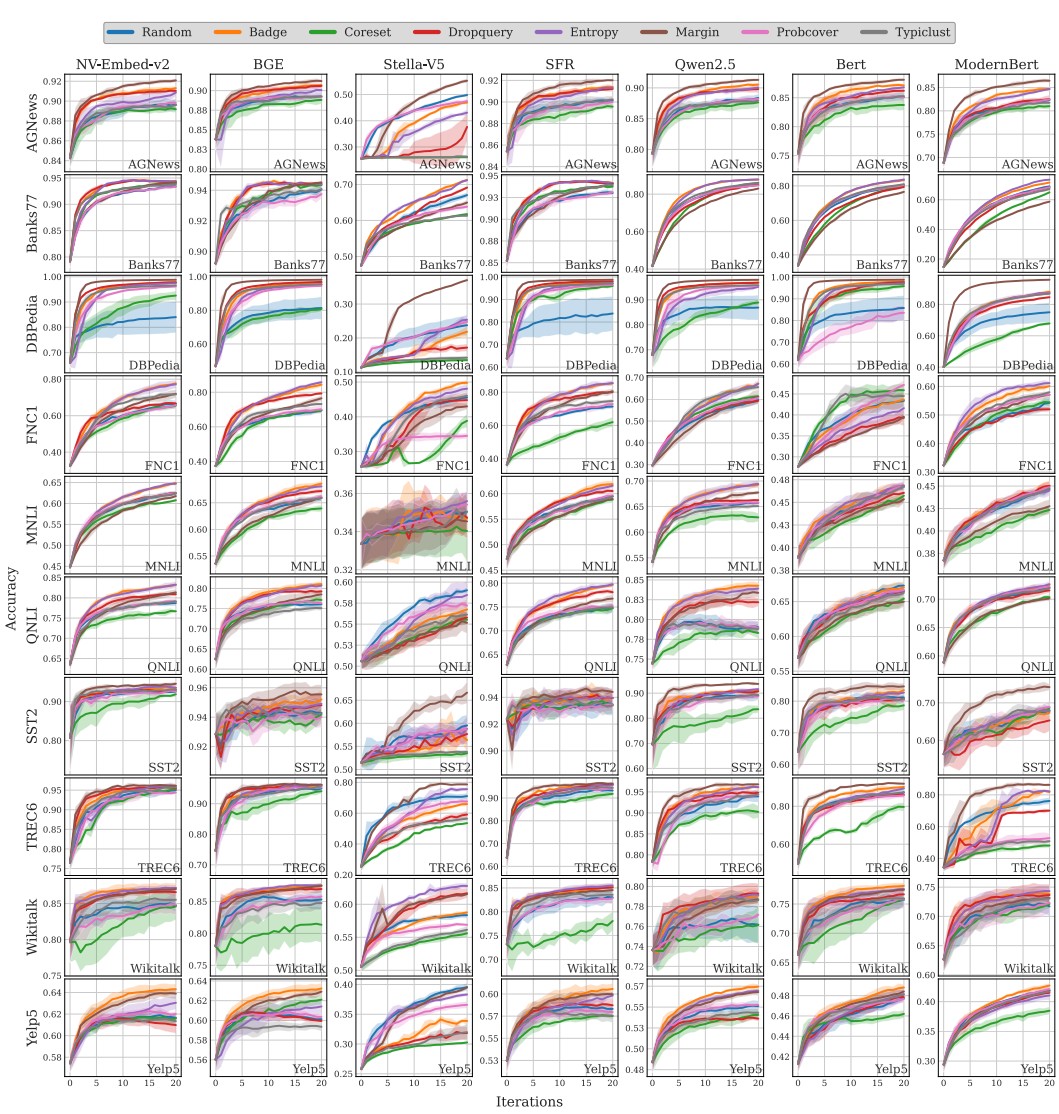

Figure 10: Performance comparison using of models and datasets over AL cycles with **TypiClust** IPS.

