# OpenReview forum: "No Free Lunch in Active Learning: LLM Embedding Quality Dictates Query Strategy Success"
_ICLR.cc/2026/Conference — Submitted to ICLR 2026_

### Official Review · Reviewer_jSqz · 2025-10-28

**Soundness:** 2
**Presentation:** 2
**Contribution:** 2
**Rating:** 4
**Confidence:** 4

**Summary:**

This paper benchmarks active learning methods that utilize large language models (LLMs) as feature extractors. The authors conduct experiments using three initial pool selection methods and seven LLMs that achieved high performance on the static embedding tasks. The main conclusions from this benchmarking are as follows: 1) Typiclust outperforms coreset and random as an initial pool selection method, 2) High-quality LLM embeddings improve active learning performance.

**Strengths:**

1. The authors investigate the performance of recent LLMs when applied to active learning. Existing benchmarks, such as MTEB (Muennighoff et al., 2023), are predominantly static benchmarks, making this the first attempt to examine LLM performance in active learning.

2. The authors demonstrate that Typiclustt (Hacohen et al., 2022) outperforms Coreset (Sener & Savarese, 2018) and random selection as an initial pool selection method.

3. The authors demonstrate that high-quality LLM embeddings like NV-EMBED-V2 and QWEN2.5 improve active learning performance compared to BERT embeddings.

**Weaknesses:**

1. The authors should conduct a more detailed and, if possible, theoretical analysis of why TypiClust is superior as an initial pool selection method. A deeper analysis would provide valuable insights for future researchers aiming to improve initial pool selection strategies.

2. The authors should provide a more in-depth analysis of why "Badge’s effectiveness is enhanced by higher-quality LLM embeddings.” Such an analysis would offer meaningful insights for future researchers seeking to advance active learning methods.

3. The right panel of Figure 7 shows that the optimal active learning method varies depending on the task. A more thorough analysis should be conducted to understand why this occurs. In particular, offering clear criteria for selecting an active learning method for new datasets would significantly enhance the practical impact of this work.

**Questions:**

Please add explanations regarding the weaknesses.

---

### Official Review · Reviewer_vZV1 · 2025-11-01

**Soundness:** 2
**Presentation:** 2
**Contribution:** 2
**Rating:** 2
**Confidence:** 4

**Summary:**

The paper presents a benchmark study on how the quality of frozen LLM embeddings and initial pool selection affect the performance of active learning strategies for text classification.

**Strengths:**

- The benchmark is technically solid and clearly described. Using frozen embeddings and a fixed classifier effectively isolates the effect of embedding quality on AL.
- The experimental design covers a reasonable space of embedders and strategies.
- The framework could be useful for future research on deep AL pipelines with LLM features.

**Weaknesses:**

- I find the paper lacking in conceptual novelty. The central takeaways that better embeddings help AL, and diversity in initial sampling complements uncertainty-based querying, are intuitive and have been reported before.
- Too few IPS strategies are tested. The results hinge heavily on IPS, yet only three methods (Random, CoreSet, TypiClust) are tested. Since CoreSet performs worse than random and TypiClust is the only one that helps, the conclusions around IPS feel narrow.
- The benchmark uses mostly standard text classification tasks, many of which saturate quickly and offer limited headroom for AL to matter. More challenging datasets (e.g., MMLU, ARC-Challenge) or multimodal extensions (vision) would make the conclusions more general. Similarly, relying solely on logistic regression as the classifier limits relevance, as many AL setups fine-tune small encoders. Overall, for a benchmark study, the experimental scope feels narrow.
- The claim that TypiClust IPS "preconditions" the learning process is vague. It seems obvious that if the initial pool is diverse, uncertainty-based strategies will appear more effective afterward. The observed effect may simply reflect complementarity between early diversity and later uncertainty, not a deeper phenomenon.
- The paper states that Badge is more robust while Margin achieves isolated spikes, but Figure 5 actually shows Margin as consistently strongest across cycles. If i read the results correctly, some parts of the discussion are misaligned with the quantitative results.
- Figure 6 is difficult to interpret; the curves are heavily bundled, and Figures 5 and 6 are referenced out of order. Presentation could be improved.

**Questions:**

1. Do you have actual correlation coefficients between MTEB rank and AL performance for different values of $k_0$?
2. Would the findings hold if the classifier were a small fine-tuned encoder or MLP head rather than logistic regression?

---

### Official Review · Reviewer_A4Kp · 2025-11-01

**Soundness:** 3
**Presentation:** 3
**Contribution:** 2
**Rating:** 4
**Confidence:** 5

**Summary:**

This paper studies whether active learning still works well under LLMs. The authors test many common AL sampling strategies on several text-classification datasets using different modern embedding models instead of training full models. They find that there is no single best AL method, that is, the strategy that works best depends heavily on the quality of the text embeddings and the dataset. A key result is that choosing diverse examples at the start (especially using TypiClust) helps, and later switching to uncertainty-based methods (like Margin or BADGE) works better.

**Strengths:**

This work gives a fresh perspective by revisiting active learning in the context of modern LLM-based representations and asking whether long-standing assumptions still hold. The authors run a well-controlled set of experiments across tasks, embedding models, and query strategies. The writing is clear and easy to follow. This paper offers useful insights for researchers and practitioners working with data-efficient learning in the LLM era.

**Weaknesses:**

1. The evaluation focuses only on text classification. Prior work shows AL behavior varies across tasks like NER, QA, etc, where uncertainty signals and data structure differ. Extending to at least one structured prediction or generative task is important.
2. While the paper convincingly shows that embedding quality affects the performance of AL strategies, the analysis remains insufficient. Should discuss which embedding properties (e.g., cluster tightness, inter-class margin structure) drive this performance.

**Questions:**

1. About TypiClust: Although TypiClust was originally proposed in a setting where representations are learned through self-supervised training, this work evaluates it in a frozen-embedding mode, where semantic structure is already strong and fixed. This design choice is reasonable for isolating embedding effects; however, it may create an implicit advantage for clustering-based diversity methods, which are particularly well-aligned with static high-quality feature spaces. Therefore, the fairness problem should be considered.
2. About noisy data: Because the study relies on frozen embeddings, the system cannot adapt to noisy or mislabeled instances. In scenarios where embeddings are fixed, noise can disproportionately influence uncertainty and clustering signals, potentially influencing AL behavior. More discussion is needed.

---

### Official Review · Reviewer_xaoE · 2025-11-01

**Soundness:** 3
**Presentation:** 3
**Contribution:** 3
**Rating:** 6
**Confidence:** 3

**Summary:**

This paper benchmarks how the quality of frozen LLM embeddings impacts the performance of deep active learning strategies for text classification. The central finding is that: the optimal active learning query strategy is not universal, but is highly dependent on the specific embedding model and the task. The study reveals that using a diversity-based strategy TypiClust for the initial pool selection provides a significant performance advantage in early AL rounds, especially when paired with high-quality embeddings. While the computationally cheap Margin sampling can perform well on specific datasets, the Badge strategy demonstrates greater robustness across tasks, and its effectiveness is notably enhanced by higher-quality embeddings.

**Strengths:**

1. This paper stuides the problem of active learning with This paper studies the problem of active learning where the success of query strategies is dictated by the quality of frozen LLM embeddings, which is a practical topic in deep active learning.
2. The paper is generally well presented with good clarity and thus it is easy to follow.
3. The experimental section is detailed, supporting a comprehensive empirical discussion.

**Weaknesses:**

1. One concern is about badge setting part. The experiments use training and test sets without a separate validation set. The badge sampling strategy as the central to the proposed method, is performed on the test dataset. This seems to maybe create a potential data leakage problem, as the badge selection mechanism may be indirectly optimizing on test data. However, the paper does not acknowledge or discuss this potential concern. If it does stand as an issue, authors are suggested to provide an discussion or analysis on whether this setup could lead to any negative effect.
2. The paper mentions "paired analyses at the seed level" but lacks clarity on the statistical testing procedure. It is unclear whether the authors are conducting paired significance testing? And the specific test used (paired t-test, Wilcoxon signed-rank test, etc) is not specified. Authors are suggested to provide clarification on this.
3. Throughout the paper, the authors mention excluding tied cases from their analysis, but critical information is missing. Why ties are excluded from the evaluation, and the total percentage or number of tied cases is not disclosed. The problem is that excluding ties could artificially inflate performance differences and bias the reported metrics with a larger win rate.
4. Several minor issues in writing, e.g., Line 215 Gwen, Line 503 Appendix 5 is confusing.

**Questions:**

Please refer to the above weakness for details.

---

### Meta-Review · Area_Chair_ahay · 2026-01-05

**Summary:**

This paper revisiting active learning (AL) in the era of LLMs, specifically focussing on the role of embedding quality, as induced by pre-trained LLMs. The results offer some practical guidance for this setting, namely that starting with a diverse set of instances is important, and switching to uncertainty sampling later on in the process is beneficial.

I think the main concern here is the scope of the contribution, which feels a little thin without some sort of theoretical work to complement the empirical findings (which, while welcome, are in line with what we might expect a priori).

**Reviewer Concerns:**

The authors do not appear to have provided a rebuttal.

**Reviewer Scores:**

No rebuttal was provided, so I'm not sure.

---

### Decision · Program_Chairs · 2026-01-26

Reject